# Factors Associated with Low Vitamin D Status among Older Adults in Kuwait

**DOI:** 10.3390/nu14163342

**Published:** 2022-08-15

**Authors:** Thurayya ALbuloshi, Ahmed M. Kamel, Jeremy P. E. Spencer

**Affiliations:** 1Hugh Sinclair Unit of Human Nutrition, Department of Food and Nutritional Sciences, School of Chemistry, Food and Pharmacy, University of Reading, Reading RG6 6AP, UK; 2Palliative Care Center, Kuwait, Ministry of Health, Al Sabah Medical Area, P.O. Box 5, Kuwait City 13001, Kuwait; 3Clinical Pharmacy Department, Faculty of Pharmacy, Cairo University|Kasr El-Aini, Cairo 11562, Egypt

**Keywords:** vitamin D deficiency, cross-sectional study, 25-hydroxyvitamin D, vitamin D, older people, Kuwait

## Abstract

Low vitamin D levels among older people represent a significant health problem worldwide. This study aimed to examine the factors associated with vitamin D deficiency in older people (aged ≥ 65) in the Kuwaiti population. A cross-sectional study was conducted in seven primary healthcare centers across Kuwait (November 2020 to June 2021). The participants (*n* = 237) had their serum vitamin D (25(OH)D) concentrations (analyzed using LC-MS) classified as sufficiency 75 nmol/L (30 ng/mL) or deficiency < 75 nmol/L (below 30 ng/mL). The data were collected using self-administered questionnaires and face-to-face interviews with participants in geriatric clinics. Binomial logistic regression analysis was applied to assess factors associated with vitamin D deficiency. Vitamin D deficiency was found to be present in two thirds of the participants (*n* = 150, 63%), with a higher prevalence of deficiency in participants who did not receive vitamin D supplements, compared to those who did (84% vs. 16%, *p* = 0.001). The results from the binary logistic regression showed that a low duration of sun exposure (OR = 0.24, 95% C.I. [0.08–0.7], *p* = 0.011), dark skin pigmentation (OR = 4.46, 95% [1.35–20.49], *p* = 0.026), and lower caloric intake (OR = 0.9, 95% C.I. [0.85–0.96], *p* = 0.001) were risk factors for vitamin D deficiency. Furthermore, a significant inverse relationship was found between vitamin D levels and parathyroid hormone (PTH) levels (OR = 1.16, 95% C.I. [1.04–1.31], *p* = 0.016). These findings support the recommendation that vitamin D supplementation and adequate sunlight exposure are necessary for raising low vitamin D levels in older people in Kuwait.

## 1. Introduction

Globally, the number of people aged 65 and older is expected to double from 2012 to 2060 [1]. Kuwait’s older population is expected to increase to 17.9% by 2050 [2]. This dramatic growth in the population could pose a challenge to public healthcare regarding vitamin deficiency in older adults because of inadequate nutrition and malnutrition [3,4]. Vitamin D deficiency is a significant public health problem worldwide, especially among older populations [5], where approximately one billion people are believed to be deficient [5], including in countries with adequate UV exposure all year round [6]. Recent research has indicated that low 25-hydroxyvitamin D (25(OH)D) concentrations are linked to diseases such as obesity, diabetes, insulin resistance, sarcopenia, immune deficiency, certain cancers, increased fall risk, mental health issues, and cardiovascular disease [7,8,9]. In contrast to other vitamins, vitamin D can be synthesized in the body in response to UV exposure, as well as through ingestion via the diet (for example, oily fish, eggs, supplementation) [10]. Despite this, vitamin D insufficiency is common, particularly at high latitudes where UV exposure is greatly reduced [11]. This cutaneous synthesis of cholecalciferol [12] is also influenced by aging, with age-dependent decreases in 7-dehydrocholesterol formation reported [13]. This is probably due to changes in skin physiology (appearance of wrinkles and thinning of the skin) which impact on vitamin D synthesis [14]. Furthermore, reductions in physical activity in older adults can reduce sun exposure considerably, thereby negatively impacting on vitamin D synthesis [15].

Screening of vitamin D has increased in recent years [16]. However, the definition and importance of low vitamin D are still unclear [17]. Serum 25-hydroxyvitamin D is supposed to be the best marker for evaluating vitamin D levels in subjects [18,19]. An individual with a 25(OH)D concentration of less than 75 nmol/L (or 30 ng/mL) is considered to be deficient in vitamin D [20,21].

Studies conducted in the Middle East, where UV exposure is largely favorable for year-round vitamin D synthesis, indicate low serum levels of vitamin D across age groups, regions, and genders [22,23,24]. However, none of these studies have addressed the issue in older populations, and there is a lack of information on vitamin D deficiency in older people in Kuwait. In Kuwait, where high temperatures (sometimes exceeding 50 °C) persist throughout the summer months, many of the inhabitants remain indoors, and outdoor exercise is rare, even over short distances of movement or travel [25]. The Kuwaiti population is also subject to several other risk factors that influence vitamin D synthesis [26] including full-body clothing, inadequate physical activity, obesity, and socio-demographic factors [27]. However, it is unclear which of these factors are most important in older adult populations. Given the accumulating data relating to the health effects of vitamin D, it is important to understand the factors associated with this deficiency in older adults [28]. In this study, the factors associated with vitamin D deficiency in older people (aged ≥ 65) were examined in Kuwait.

## 2. Methods and Materials

### 2.1. Population and Study Design

This study was conducted from November 2020 to June 2021, taking place during the COVID-19 pandemic. Therefore, the researchers followed regulations, instructions, and guidelines of the Ministry of Health and World Health Organization (WHO), in respect of preventing the spread of COVID-19, such as wearing masks, physical distancing, and sterilization after interviewing each participant. Participants were recruited from seven primary healthcare centers providing geriatric clinics [29]. However, only four older adults were recruited per day to avoid crowds and close contact.

The protocol of this research was reviewed and approved by the Kuwait Ministry of Health Standing Committee for the Co-ordination of Medical Research (approval protocol no: 2019/1016) and by the University of Reading Ethics Committee for Clinical Research (approval protocol no.: URCE 19/47). The principles of the Declaration of Helsinki were followed by this project. Once the trial’s purpose and procedures had been explained individually to each participant, consent forms were signed. The risks and benefits of the study were clearly explained in the consent form signed by anyone willing to participate.

Volunteers were appropriate for this research if they were Kuwaiti of both genders, were 65 years old or over, and had no symptoms of COVID-19. Individuals with a disability, those unwilling to join, and those with any illness that might influence 25[OH]D synthesis (for example, skin cancer, kidney disease, or liver disease), as well as volunteers infected with COVID-19, were excluded from this study. Moreover, only participants with complete data for measured 25[OH]D and another required laboratory test (serum calcium, phosphorus, PTH, and alkaline phosphatase (ALP)) were included in the analyses. The entire questionnaire was validated and piloted with ten volunteers not included in the main research project. The sample size was 237 participants [29].

### 2.2. Collection of Data

Research information, including convenience blood tests, was collected through face-to-face interviews with participants and from patients’ registries and medical records using questionnaires. The Ministry of Health was requested to spread the study and invitation poster in their electronic system across primary healthcare centers that were providing geriatric clinics and place the poster at each clinic’s entrance to encourage older people to participate in this study. Those participants who were willing to enroll in the trial could register by giving their details to the researcher or other team members (geriatric physician).

#### 2.2.1. Demographic Variables

The independent variables included marital status, age, monthly income and occupation, and type of accommodation. The participants were also asked how many children they had and about their educational level.

#### 2.2.2. Lifestyle Variables and Sun Exposure

The lifestyle variables, including smoking status and alcohol consumption, were classified into ‘Yes’/‘No’ categories, whereas sleep duration (day/h) was classified into three groups: ≥9 h, 7–8 h, and ≤6 h. Physical activity was assessed with the International Physical Activity Questionnaire for the Elderly (IPAQ) [30], due to its previously acceptable validity and reliability in Arabic [31]. To assess participants’ attitudes toward sun exposure, a questionnaire was administered, asking them to recall how long they were exposed to the sun each day for one week and whether or not they used sun cream, with Yes/No responses [32]. Here, the participants could respond by choosing from among three durations of sun exposure: 0 = ≤ 5 min, 1 = 5–30 min, and 2 = ≥30 min [32]. There were also three possible responses for the type of clothing worn, and the parts of the body exposed to the sun. This was ascertained by asking the participants to indicate the diagram that most closely matched their style of dress (1, 2, or 3). The female participants were classified according to three diagrams of this nature, namely, Picture 1: hijab (puberty; face and hands exposed); Picture 2: veiled (black veil; most of the face and body covered); and Picture 3: without hijab (wearing Western-style clothing). Conversely, the male participants were classified according to three diagrams based on their style of dress, namely, Picture 1: dishdasha (a long-sleeved, collarless garment) and ghutra (a traditional headdress), which is a mode of dress that is usually adopted at puberty, leaving the face and hands exposed; Picture 2: dishdasha without ghutra (adopted at puberty, leaving the face, head, and hands exposed); and Picture 3: Western dress, for example, a cap and trousers. Additionally, one assessor used the ‘Fitzpatrick Classification of Skin Phototype’ to determine the participants’ skin colors as (I) very fair, (II) fair, (III) fair to medium, (IV) medium, (V) olive or dark, and (VI) very dark with deep pigmentation [29]. Skin types (VI) and (I) were not found in this sample.

#### 2.2.3. Anthropometric Measurements

Anthropomorphic measurements comprised body mass index (BMI), which was categorized as reported by the WHO cut-off points for normal weight (18.5–24.9 kg/m^2^), overweight (25.0–29.9 kg/m^2^), and obese (30 ≥ kg/m^2^) [33]. In addition, the WHR was calculated by measuring WC at the level of the umbilicus, using a non-stretch measuring tape, and with the participant wearing light clothes. Meanwhile, hip circumference was determined as the widest diameter of the hip. According to the WHO recommendations, the cut-off values for an elevated WHR were ≥0.85 for females and ≥0.90 for males [33]. Moreover, WCs of ≥80 cm in women and ≥94 cm in men were considered high [34,35].

#### 2.2.4. Vitamin D Intake, Dietary Calcium, and Calories

Information about vitamin D, calcium, daily calorie intake (Kcals), and supplementation of calcium and vitamin D was gathered using the Food Frequency Questionnaire (SFFQ). This represented a version of a previously validated questionnaire [29,36,37]. The questionnaire was translated from English into Arabic. In addition, some items listed in the SFFQ are sources of natural vitamin D (such as eggs, poultry liver, meat, fish, and tuna). To ensure that the older participants understood this, the researcher explained the list of food items in the SFFQ, using photographs and food portion sizes via a food model, measured with an electronic kitchen scale. NutriBase Pro 19 is a software program used to analyze the content of foods within the meals selected by the volunteers. Because traditional Kuwaiti dishes are not available in NutriBase, the ‘Food composition database: Kuwaiti composite dishes’ was used [38]. In addition, food labels were referred to when deriving the vitamin D values of local food and supplements, for example, a local buttermilk called ‘laban’ and vitamin D-fortified milk and yogurt. The Arabic version of the SFFQ was piloted with older adults (*n* = 10) to ensure that all the terms were easy to comprehend. Furthermore, this instrument is suitable for assessing vitamin D and calcium intake.

#### 2.2.5. Clinical Variables

Participants’ clinical variables were also obtained, such as any previous diagnosis of chronic diseases (for example, cardiovascular disease (CVD), type 2 diabetes, dyslipidemia, osteoporosis, and hypertension), and all the volunteers were referred by medical physicians who physically examined them to confirm previous diagnoses. The investigators had access to participants’ records of medical history. A mercury sphygmomanometer was used to measured blood pressure in the left upper arm [29]. The geriatrician followed the International Society of Hypertension Global Hypertension Practice Guidelines to measure blood [39].

#### 2.2.6. Biochemical Assessment

##### Assessment of the Biochemistry Test, 25(OH)D, and PTH

A nurse collected 10 mL of blood from every volunteer and placed each test in a gel tube (BD Vacutainer, SST II Advance). These samples were then protected from the light. The samples were centrifuged at 2000× *g* for 15 min (on the collection day), before pouring the serum into Eppendorf tubes to be stored at −80 °C until the samples were required for analysis [29]. The serum 25(OH)D concentration was adopted to measure the vitamin D level because this is the main circulating form of vitamin D, composed of vitamin D ingested in food and through sun exposure [40]. The collected fasting blood samples were analyzed in a College of American Pathologists-accredited laboratory using liquid chromatography-tandem mass spectrometry (MC/MS/MS). This method is recommended for assessing vitamin D status in epidemiological studies [29].

Following the Endocrine Society guidelines, the following cut-off for 25(OH)D was determined to define vitamin D sufficiency in this study, ≥75 nmol/L (30 ng/mL), whereas vitamin D insufficiency was determined as 50–75 nmol/L (20–30 ng/mL), and vitamin D deficiency as <50 nmol/L (20 ng/mL) [21]. Therefore, in the present study, serum 25(OH)D levels < 75 nmol/L (below 30 ng/mL) were considered to pose a risk of 25(OH)D deficiency, in accordance with the Institute of Medicine of the United States National Academy of Sciences, and the US Endocrine Society [17,21,40]. Moreover, serum intact parathyroid hormone was measured using the Access Intact PTH electrochemiluminescence immunoassay with a commercial kit version of the Cobas E601 module analyzer, according to the manufacturer’s instructions. The reference PTH range is 1.6–6.9 pmol/L for normocalcemic patients. Furthermore, phosphorus (serum mmol/L) levels were measured with phosphomolybdate UV (Architect plus c 4000, Abbott, Abbott Park, IL, USA), as was alkaline phosphatase U/L. In addition, serum calcium was measured using Arsenazo III dye (Architect plus c 4000, Abbott, Abbott Park, IL, USA). The internal and external methods mentioned above were validated to ensure that accurate and precise results were reported.

### 2.3. Statistical Analysis

Data analysis was performed using the Statistical Package for Social Sciences (SPSS) v 26, R v 4.2. Categorical data were summarized using counts, and percentages and continuous data were summarized using the mean ± standard deviation for continuous normal variables and the median/interquartile range (IQR) for non-normal variables. A chi-square test of independence (χ^2^) was conducted to assess the association between vitamin D status (deficiency; sufficiency) and categorical socio-demographic characteristics/lab data of the included respondents. An unpaired *t*-test was likewise carried out for the continuous normal variables, and the Mann–Whitney test for the non-normal variables. Binary logistic regression was used to assess the independent factors associated with vitamin D deficiency. Meanwhile, the odds ratio (OR) was applied as a measure of effect size, and the corresponding 95% confidence interval (CI) was used to test the hypothesis. The variables associated with vitamin D deficiency in the univariate analysis were included in the model. The initial variables included were age, gender, daily calorie intake (Kcals), PTH level, vitamin D intake, calcium intake, diabetes, cardiovascular diseases, body mass index (BMI) category, sun exposure, seasonality, and pigmentary phototype. Backward stepwise elimination was then performed to reduce the complexity of the model, based on the AIC, until the final model was obtained. Pigmentary phototypes II and III were then combined to ensure an adequate sample size. A *p*-value < 0.05 was considered statistically significant.

## 3. Results

A total of 237 participants aged 65 years and over were included in this study. Participants were categorized based on their 25(OH)D values, falling into two categories: deficiency (<75) and sufficiency (75–250). Table 1 shows the study population’s socio-demographic characteristics, clinical data, and laboratory parameters, grouped by vitamin D category (sufficiency; deficiency) (Appendix A). In the vitamin D deficiency group, 51% of the participants were males and 49% were females, compared to 37% and 63% in the vitamin D sufficiency group. The mean age of the vitamin D deficiency group was 71.5 ± 5.2 years compared to 71.2 ± 4.9 years in the vitamin D sufficiency group.

Additionally, the mean serum vitamin D levels were higher in participants with vitamin D sufficiency (105.2 ± 35.9) compared to participants with a deficiency of vitamin D (49.3 ± 14.8). Parathyroid levels were higher in participants with vitamin D deficiency (6.03, IQR 4.40; 7.82) than in participants with vitamin D sufficiency (5, IQR 3.67; 6.50). Data from the completed Semi-quantitative Food Frequency Questionnaire (SFFQ) revealed that a greater number of participants in the sufficiency group consumed vitamin D supplements (*n* = 84, 84.0%), compared to the deficiency group (*n* = 16, 16.0%), as illustrated in Figure 1. Moreover, dietary calcium intake was significantly higher among the participants in the sufficiency group (855.29, IQR 639.77; 1124.40) than in the deficiency group (621.34, IQR 458.94; 943.24). Additionally, daily caloric intake was significantly higher in the participants with vitamin D sufficiency (1567.0 ± 456.61) than in the deficiency group (1337.49 ± 497.74).

Table 2 shows the study population’s comorbidities, laboratory parameters, sun exposure behavior, seasonality, and skin pigmentation, categorized according to vitamin D status (sufficiency or deficiency). A significant difference between the groups was observed in terms of the pigmentary phototype. In the vitamin D sufficiency group, 34% of the participants belonged to the Type III category, 62% belonged to the Type IV category, and 3% belonged to the Type V category of pigmentary phototypes. Meanwhile, in the vitamin D deficiency group, 19% belonged to the Type III category, 67% belonged to the Type IV category, and 13% belonged to the Type V category of pigmentary phototypes. Furthermore, a significant association was observed between sun exposure and vitamin D status. In the vitamin D deficiency group, 95% of the participants reported sun exposure of <5 min/day compared to 85% of the participants in the vitamin D sufficiency group (*p* = 0.004).

### Independent Factors Associated with Vitamin D Deficiency

Binary logistic regression was performed to check the factors associated with vitamin D for all the participants (Figure 2 and Appendix A). Gender, sun exposure period, calorie intake, pigmentation phenotype, and PTH were found to be independent predictors of vitamin D deficiency. The results show that vitamin D deficiency was more likely among the male than among the female participants (OR = 2.13, 95% C.I. [1.16–4.02, *p* = 0.017). Moreover, sun exposure showed a positive association with vitamin D deficiency. Respondents who were exposed to sunlight for 5 to 30 min per day were less likely to show vitamin D deficiency than participants who were exposed to sunlight for less than 5 min daily (OR = 0.24, 95% C.I. [0.08–0.7], *p* = 0.011). Pigmentation phenotype V (dark skin) showed a negative association with vitamin D deficiency (OR = 4.46, 95% [1.35–20.49], *p* = 0.026). In addition, higher daily caloric intake (100 Kcal reduction) was negatively associated with vitamin D levels (OR = 0.9, 95% C.I. [0.85–0.96], *p* = 0.001); as daily caloric intake increased, the likelihood of vitamin D deficiency decreased. Additionally, higher PTH levels were associated with a greater risk of vitamin D deficiency (OR = 1.16, 95% C.I. [1.04–1.31], *p* = 0.016). The results did not change when vitamin D and calcium intake were included in the model (Appendix A).

## 4. Discussion

This research investigated factors associated with vitamin D deficiency in older people aged 65 years and over. Vitamin D deficiency was found to be common in the population, with males having lower vitamin D levels relative to females. The main factors related to having a low plasma vitamin D status were observed to be the absence of vitamin D supplementation, low calorie intake, low sun exposure, and dark skin pigmentation. In contrast, physical activity, sleep duration, comorbidities, smoking status, alcohol consumption, obesity, seasonality, and style of dress did not affect 25(OH)D levels in the sampled older people. Additionally, parathyroid hormone (PTH) levels were found to be elevated due to vitamin D deficiency. Different opinions exist regarding the optimal serum 25(OH)D concentration, with the Endocrine Society recommending 25(OH)D concentrations greater than 75 nmol/L [22], whereas the Institute of Medicine (IOM) [41], EFSA, and European Calcified Tissue Society [42] suggest a serum 25(OH)D level of over 50 nmol/L. In this paper, vitamin D deficiency was defined as a serum 25(OH)D level below 75 nmol/L, with 63% of the older participants found to be deficient. High levels of deficiency have previously been reported in older people of other Middle Eastern countries, with a prevalence of <20 ng/mL in 38–77% in older populations in Egypt, Iran, and the Lebanon, and 65% in the Kingdom of Saudi Arabia (KSA), with a range of 0–28 ng/mL [43]. Elsewhere, deficiencies have been reported in Germany at 54.5% (age group 18–79 years) [44], the US at 41.1% (≥65 years old) [28], Europe at 40.4% [45], China at 34.3% (>60 years old) [46], and France at 34.6% (18–89 years) [47]. Conversely, a somewhat lower prevalence (22.7%) was observed in the older population in Australia [48], possibly due to a combination of climate and a greater culture of outdoor living.

The current study identified vitamin D supplementation as a significant predictor of vitamin D status. Moreover, it is possibly more efficacious in raising serum 25(OH)D concentrations in a senior population [49], where there can be a tendency towards insufficient sunlight exposure and low dietary intake of vitamin D [21]. Nevertheless, studies indicate that the dose should be precisely defined to achieve optimal serum 25(OH)D levels in older populations [50]. The Endocrine Society’s Clinical Practice Guidelines recommend that all adults who are deficient in vitamin D be treated with 50,000 IU of vitamin D3 or D2 once a week for eight weeks—or its equivalent, 6000 IU of vitamin D2 or D3 daily—in order to raise 25(OH)D concentrations to above 30 ng/mL, followed by maintenance therapy of 1500–2000 IU/d [21]. It was also found that vitamin D deficiency is slightly more common in male participants (51%) compared to females (49.0%), which may be explained by the fact that more women consume vitamin D supplements (65%) compared to men (35.0%). Other studies report mixed gender-related results, with some reporting a higher prevalence of vitamin D deficiency among men [51,52], and others indicating that older women have lower 25(OH)D levels than older men [53,54,55].

Unsurprisingly, it was also observed that dark skin pigmentation is associated with vitamin D deficiency. In the current study, most of the elderly population had a dark to medium skin color (IV 67% and V 13%). Previous studies have emphasized that dark skin inhibits cutaneous pre-vitamin D synthesis [56], especially in Asians and African Americans, who have the lowest vitamin D levels [57,58,59]. The current study also showed that participants who were exposed to sunlight for 5–30 min per day were less likely to display vitamin D deficiency than participants who were exposed to sunlight for less than 5 min per day. Older people are considered at high risk of vitamin D deficiency due to a number of reasons, including lack of outdoor exercise, age-related changes in UVB absorption, and the skin’s ability to synthesize vitamin D, as well as diminished expression of vitamin D receptors in the body and reduced kidney function [60]. Researchers found that aging can reduce the skin’s ability to produce pre-vitamin D3 by as much as half when comparing the amount produced by young adults [61]. Previous research reported that individuals with darker skin would need 10 times as much sun exposure as those with lighter skin to obtain the same amount of vitamin D3 [62]. Seasonality was considered in this study but not identified as being significantly associated with 25(OH)D. Despite the seasonal variation reported in the region, low vitamin D is more prevalent in the summer than in the winter [63,64]. In Kuwait, sun exposure is avoided in the desert climate, especially when temperatures are extremely high [40].

With respect to nutritional factors, it was observed that a lower daily caloric intake was negatively associated with vitamin D levels. This observation is unsurprising based on the fact that dietary sources of cholecalciferol (vitamin D3) usually consist of calorie-dense foods such as eggs, fatty fish—such as salmon, sardines, mackerel, and tuna—and some fortified cereals, juices, and milk. Previous evidence has confirmed that the prevalence of vitamin D deficiency in older adults is associated with insufficient daily intake of calories and calcium (all *p* < 0.05) [3]. Dietary intake of vitamin D was found to be insignificant, at around 225 IU per day, which is below the recommendations set out in the Endocrine Society’s Clinical Practice Guidelines. For example, among subjects aged 19–70 years, 600–2000 IU/day of oral vitamin D is recommended, and for subjects aged > 70 years, 800–2000 IU/day of oral vitamin D is considered adequate [21,65]. For the participants in this study, oily fish consumption was low, and the participants did not consume any cereals fortified with calcium or vitamin D. Moreover, they consumed less or no milk. However, evidence suggests that dietary vitamin D constitutes < 10% of the total amount of vitamin D in the body [62], meaning that a varied and balanced diet alone is unlikely to keep plasma vitamin D levels optimal. As such, vitamin D supplementation is advised in older subjects with vitamin D deficiency, especially if they also lack sufficient solar UV exposure.

This study showed that vitamin D levels are inversely associated with PTH levels. Many older adults with low vitamin D levels present with evidence of secondary hyperparathyroidism [66,67,68]. Low 25(OH)D levels decrease intestinal calcium absorption efficiency, and the body responds by increasing PTH secretion [69]. An increased serum parathyroid hormone (PTH) concentration in older adults can cause defects in mineralization, bone loss, and bone turnover, and an increased risk of fractures [70]. Studies observed that a value of 38 ng/mL of vitamin D may be sufficient to avoid an increase in PTH [71,72], and vitamin D supplementation with calcidiol improved serum 25(OH)D while significantly lowering PTH levels and reducing secondary hyperparathyroidism [73].

The results indicated no association between BMI and 25(OH)D levels. This is in agreement with some studies in older adults [50,74]. However, other studies have shown that obesity is related to vitamin D deficiency [75,76,77,78]. In this regard, it is hypothesized that accumulated adipose tissue might absorb and sequester vitamin D, which is fat-soluble, thereby reducing the 25(OH)D circulating in the body [79,80,81]. Additionally, research findings in the literature support that obesity could be a consequence of a low vitamin D status [76]. It has also been indicated that the adipocyte-derived hormone leptin might stimulate a pathway that inhibits renal production of the active form of vitamin D [82]. The finding in this study may be due to the overall high BMI among the older people participating in this study, who were categorized as obese (41.2%), overweight (45.9%), and normal weight (12.9%). Unfortunately, the relationship between obesity and low vitamin D among older populations is not yet fully understood. Therefore, further research is necessary to clarify the causal correlation and direction between obesity and vitamin D status among older adults.

### Strengths and Limitations

In this study, vitamin D status was measured, for the first time, in a nationally representative sample of older adults in Kuwait. In order to assess plasma vitamin D levels, LC/MS/MS was used, this being considered the gold standard measurement. However, this study also has potential limitations due to its cross-sectional analysis, making it impossible to infer causality. Another factor is perhaps the lockdown in Kuwait, which began in April 2020 due to the COVID-19 pandemic. This situation made it challenging to recruit older people, out of fear of spreading the virus. However, the researchers followed the instructions and rules of the Ministry of Health and WHO to prevent the spread of the disease. The lockdown itself may also have increased the risk of vitamin D deficiency and its prevalence, since the primary source of vitamin D is sunlight, whereas the lockdown meant the whole population spent more time indoors. In addition, information about the participants’ attitudes to sun exposure was assessed using a questionnaire. In this case, older people might not provide accurate information. Therefore, it would be necessary to monitor the solar UV intensity and cumulative dose using a sensor, such as the Solarmeter 6.5. This is a simple, hand-held biometer that measures changes in UVR levels in different environmental conditions with relative accuracy and user-friendliness [73] for future intervention studies. Meanwhile, the anthropometric measurements included were BMI and WHR, in accordance with the WHO. However, in this study, we did not measure the lost muscle mass as the calf circumference, which is considered to be the most appropriate evaluation in older people. Finally, the researcher did not mention the medications that could impair vitamin D sufficiency, such as cholesterol-lowering medicines (for example, colestipol and cholestyramine).

## 5. Conclusions

In this study, vitamin D deficiency was noted in 63% of older people in a population sample. Moreover, 25(OH)D levels were identified as significantly associated with the male gender, low sun exposure, dark skin pigmentation, and low-calorie intake. Moreover, elevated PTH was found as a result of vitamin D deficiency. This was found despite the study being conducted at a latitude where adequate sunlight is available throughout the year. Therefore, vitamin D supplements and adequate sun exposure are crucial factors in avoiding low 25(OH)D levels. Screening vitamin D status and running awareness programs to develop adequate vitamin D intake are therefore two strategies proposed for avoiding a low vitamin D status among older people.

## Figures and Tables

**Figure 1 nutrients-14-03342-f001:**
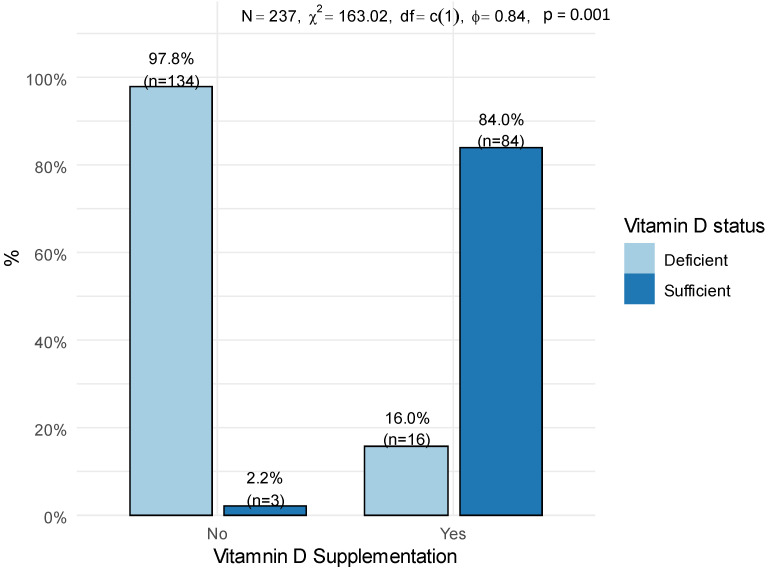
Association between vitamin D supplementation and vitamin D deficiency. Statistical analysis was performed using a chi-square test of independence. N, number of participants; χ^2^, chi-square statistic; ϕ, phi coefficient.

**Figure 2 nutrients-14-03342-f002:**
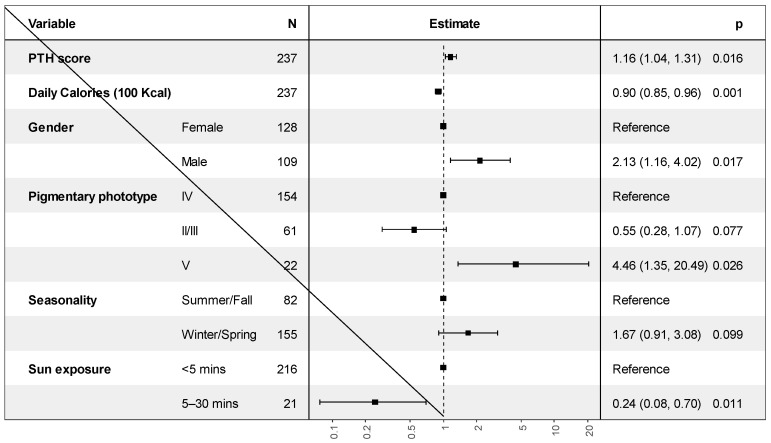
Results of binomial logistic regression analysis for the factors associated with vitamin D deficiency.

**Table 1 nutrients-14-03342-t001:** Socio-demographic characteristics, lifestyle variables, and serum 25-hydroxyvitamin D concentration in a senior population, with vitamin D status.

Variables	Deficiency (<75) (*n* = 150, 63%)	Sufficiency (≥75) (*n* = 87, 37%)	*p*-Value
Serum vitamin D levels (SD) (nmol/L)	49.25 (14.81)	105.24 (35.93)	<0.001
Age in years, mean (SD)	71.52 (5.15)	71.21 (4.56)	0.630
Gender			0.042
Female	73 (48.67%)	55 (63.22%)	
Male	77 (51.33%)	32 (36.78%)	
Marital status			0.595
Divorced	5 (3.33%)	3 (3.45%)	
Married	105 (70.00%)	68 (78.16%)	
Single	3 (2.00%)	1 (1.15%)	
Widowed	37 (24.67%)	15 (17.24%)	
Governorate			0.126
Ahmadi	19 (12.67%)	14 (16.09%)	
Capital	24 (16.00%)	13 (14.94%)	
Farwanya	25 (16.67%)	14 (16.09%)	
Hawaly	27 (18.00%)	23 (26.44%)	
Jahra	23 (15.33%)	16 (18.39%)	
Mubarak Al-Kabeer	32 (21.33%)	7 (8.05%)	
Type of House			0.116
Rental flat	2 (1.33%)	1 (1.15%)	
Rental house	0 (0.00%)	2 (2.30%)	
Owned flat	0 (0.00%)	1 (1.15%)	
Owned house	148 (98.67%)	83 (95.40%)	
Education level			
No formal education	31 (20.67%)	20 (22.99%)	0.728
Completed primary/intermediate school	26 (17.33%)	10 (11.49%)	
Completed secondary school	36 (24.00%)	19 (21.84%)	
Completed diploma	25 (16.67%)	18 (20.69%)	
University degree or above	32 (21.33%)	20 (22.99%)	
Income per month			0.691
KWD 500–1000	62 (42.18%)	42 (48.84%)	
KWD 1001–1500	37 (25.17%)	22 (25.58%)	
KWD 1501–2000	27 (18.37%)	12 (13.95%)	
More than KWD 2000	21 (14.29%)	10 (11.63%)	
Occupation			0.658
Business	8 (5.33%)	2 (2.30%)	
Housewife	44 (29.33%)	29 (33.33%)	
Paid job (with salary)	4 (2.67%)	3 (3.45%)	
Retired	94 (62.67%)	53 (60.92%)	
Number of children	6.00 [5.00; 9.00]	6.00 [4.00; 9.00]	0.193
BMI (Kg/m^2^)	29.63 (6.14)	30.31 (5.82)	0.400
Body mass index categories:			0.297
Normal weight	17 (11.33%)	14 (16.09%)	
Overweight	75 (50.00%)	35 (40.23%)	
Obese	58 (38.67%)	38 (43.68%)	
Waist (CM)	100.69 (19.30)	103.19 (17.55)	0.310
Hip (CM)	106.29 (19.08)	109.66 (13.79)	0.118
Current smoker	11 (7.33%)	6 (6.90%)	0.900
Alcohol drinker	2 (1.33%)	2 (2.30%)	0.626
Sleep duration category			0.348
<6	28 (18.92%)	12 (13.79%)	
6–8	118 (79.73%)	72 (82.76%)	
>8	2 (1.35%)	3 (3.45%)	
Vitamin D supplement consumers	16.0 (16.0%)	84.0 (84.0%)	<0.001
Calcium supplementation (%)	3 (2.00%)	4 (4.60%)	0.265
Dietary intake of vitamin D (IU) *	177.99 [98.40; 235.36]	217.35 [153.44; 307.65]	0.001
Dietary intake of calcium (mg) *	621.34 [458.94; 943.24]	855.29 [639.77; 1125.40]	<0.001
Daily calorie intake (Kcal)	1337.49 (497.74)	1567.01 (456.61)	<0.001
Walking per minute	3.25 (3.05)	2.90 (3.04)	0.385
Physical activity > 1 day	67 (44.67%)	41 (47.13%)	0.817
Dresses for women (1, 2, 3)			0.939
Hijab	34 (46.58%)	27 (50.00%)	
Veiled	37 (50.68%)	26 (48.15%)	
Without hijab	2 (2.74%)	1 (1.85%)	
Dresses for men (1, 2, 3)			0.167
Dishdasha and ghutra	74 (96.10%)	29 (87.88%)	
Dishdasha without ghutra	2 (2.60%)	3 (9.09%)	
Cap and trousers	1 (1.30%)	1 (3.03%)	

The continuous variables are expressed as the mean ± standard deviation (SD) or median [IQR]; the categorical variables are expressed as the number (*n*) and percentage (%). Statistical analysis was performed using an unpaired *t*-test for normal variables, and a Mann–Whitney test for non-normal variables. The association between categorical variables was assessed using a chi-square test of independence. * Kuwaiti dinars. * Dietary vitamin D intake (IU/day per 1000 Kcal). * Dietary intake of calcium (mg/day per 1000 Kcal). Abbreviations: 25(OH)D—total serum 25-hydroxyvitamin D; BMI—body mass index, calculated as weight in kilograms divided by height in meters squared.

**Table 2 nutrients-14-03342-t002:** Association between comorbidities, laboratory parameters, sun exposure behavior, seasonality, skin pigmentation, and vitamin D status.

Variables	Deficiency (<75) (*n* = 150, 63%)	Sufficiency (≥75) (*n* = 87, 37%)	*p*-Value
Comorbidity indicators			
Dyslipidemia	109 (72.67%)	61 (70.11%)	0.787
Hypertension	107 (71.33%)	58 (66.67%)	0.544
Type 2 diabetes	92 (61.33%)	60 (68.97%)	0.238
Cardiovascular disease	34 (22.82%)	20 (22.99%)	0.976
Osteoporosis	33 (22.00%)	26 (29.89%)	0.231
Laboratory test			
PO4 mmol/L	1.10 [1.02; 1.22]	1.15 [1.03; 1.25]	0.050
Ca mmol/L	2.29 [2.22; 2.37]	2.32 [2.26; 2.38]	0.034
PTH mmol/L	6.03 [4.40; 7.82]	5.02 [3.67; 6.50]	0.003
ALP IU/L	69.00 [57.25; 81.75]	69.00 [57.00; 90.50]	0.202
Seasonality			0.126
Winter/Spring	104 (69.33%)	51 (58.62%)	
Summer/Fall	46 (30.67%)	36 (41.38%)	
Sun exposure			0.023
<5 min	142 (94.67%)	74 (85.06%)	
5–30 min	8 (5.33%)	13 (14.94%)	
Pigmentary phototype			0.004
II	3 (2.00%)	0 (0.00%)	
III	28 (18.67%)	30 (34.48%)	
IV	100 (66.67%)	54 (62.07%)	
V	19 (12.67%)	3 (3.45%)	

The continuous variables are expressed as the mean ± standard deviation (SD) or median [IQR]; the categorical variables are expressed as the number (*n*) and percentage. Statistical analysis was performed using an unpaired *t*-test for normal variables, and a Mann–Whitney test for non-normal variables. The association between categorical variables was assessed using a chi-square test of independence. Pigmentary phototypes: (II) fair, (III) fair to medium, (IV) medium, and (V) olive or dark. Abbreviations: ALP—alkaline phosphatase; Ca—calcium; PTH—parathyroid hormone; PO4—phosphate.

## Data Availability

The data in this research are available from the corresponding author upon request.

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
