# Peer review of "Factors Associated with Low Vitamin D Status among Older Adults in Kuwait"

_nutrients, 2022, doi:10.3390/nu14163342_

Round 1

Reviewer 1 Report

There are many problems needed to revise, the detail comments are as followed:

1. Fig.1 vitamin D(yes/no), What does it mean? The specific contents and units shall be marked on the ordinate.

2. As for Table 1, there are too many mistakes. It is recommended to keep three decimal places for the p value, and the frequency of gender should be an integer. The decimal places of the percentage are recommended to be unified. For some variables, the sum of the percentages of each subgroup distribution is not 100, and the authors are advised to check.

3. The title of Figure 1 is inappropriate, and here the authors are not showing associations, but different distributions of different groups.

4. I doubt the reliability of the results of binominal logistic regression. First, are the subgroup classifications of the covariates included in the regression consistent with those presented in Table 1? The composition ratio of some subgroups is so low that they can be combined for analysis. Secondly, what are the covariates of the model, the author did not express clearly.

5. The conclusion is too long and needs to be refined.

Author Response

thanks, please see the attachment 

Reviewer 2 Report

I appreciate the opportunity to review this article.

 It is an important theme for the area and I present some considerations.

§  Title: “Factors associated with vitamin D deficiency among the elderly population in Kuwait: a cross-sectional study”. I suggest a title that is more declarative expressing the main result of the study instead of a descriptive and neutral style. 

Also, “elderly” can be substituted by “older people”. “Elderly” connotes frailty. “Older person” or “older people” are possibly preferable terms because they reflect better how the general population refers to older members of our families and communities. This change should be considered in the entire manuscript.

§  Abstract: This section needs reformulation including background, aim, methods, and results, and the conclusion in a more organized format including only the most important information.

§  Introduction: There is a predominance of general aspects of vitamin D and epidemiological data (first two paragraphs), in detriment of the study's focus on vitamin D status in older people, the question of the study. Furthermore, there is an excessive number of references. Some emphasis needs actualization, e.g., this statement had eleven years: “the percentage of older adults suffering from vitamin D deficiency ranges from 20% to 100% in the US [24]”. (line 58)

§  Methods:

The study with a vulnerable population was carried out during the COVID-19 pandemic. This fact was not contextualized in the manuscript. Could you explain how the operation took place? Were there sample losses because of this fact? Did any participants have the infection and remain in the study? 

This section should be rewritten with a fusion of some subitems, e.g., 2.1 and 2.2, and rename others most adequately. The Laboratory Methods section describes vitamin D and PTH analyses, but its name does not reflect this content. 

To ensure quality control in plasma vitamin D analysis, it is strongly recommended that details on the liquid chromatography-tandem mass spectrometry be provided in the description of methods (sample preparation and processing, equipment, and internal and external validation of the method). Unfortunately, the references cited do not clarify these aspects.

(Albolushi T, Bouhaimed M, Spencer J. Lower Blood Vitamin D Levels Are Associated with Depressive Symptoms in a Population of Older Adults in Kuwait: A Cross-Sectional Study. Nutrients. 2022 Apr 8;14(8):1548). 

Two strong weaknesses of the study concern unclear methodology of the assessment of food consumption and the data of polypharmacy that is part of the health care of the elderly. . How do the authors explain the use of a validated FFQ for pregnant women and female students to be appropriate for the older population? Furthermore, was this instrument fitted for assessing vitamin D and calcium intake? In addition to this distortion, it is not explained how the adaptations and the analysis of nutrients, software, etc. were made. Unfortunately, the references cited do not clarify these aspects [32, 39, 40]. 

-Albolushi T, Bouhaimed M, Spencer J. Lower Blood Vitamin D Levels Are Associated with Depressive Symptoms in a Population of Older Adults in Kuwait: A Cross-Sectional Study. Nutrients. 2022 Apr 8;14(8):1548.  

-Taylor C., Lamparello B., Kruczek K., Anderson E. J., Hubbard J., Misra M. Validation of a food frequency questionnaire for calcium and vitamin D intake in adolescent girls with anorexia nervosa. Journal of the American Dietetic -Association. 2009;109(3):479–485. (inside ref 32)

Papandreou D., Rachaniotis N., Lari M., Almussabi W. Validation of a food frequency questionnaire for vitamin D and calcium intake in healthy female college students. Food and Nutrition Sciences. 2014;5(21):2048–2052.

“A sun exposure questionnaire was administered to assess the participants’ attitudes towards sun exposure (e.g., sun exposure time, sun cream use, and clothing worn) [35].”Explain how the results are produced in a range of scores. 

Anthropometric measurements: the most appropriate for evaluation in older people are measurements of muscle mass lost as calf circumference. Why didn´t it was applied?

§  Results:

What were the continuous variables with asymmetric distribution if they are all presented as mean and SD?

In table 1 the weight and height data are unnecessary. In Table 2 and Figure 2, a mistake can be seen in the description of the tool chosen to assess sun exposure, as well as in the presentation of the results. Some dimensions must be minimally explained for a better understanding of the reader and reproducibility of the methods. Representative results of this measure are presented in scores ranging from zero (no exposure) to 56 (maximum exposure sure). Therefore, statistical associations are weakened.

The results related to "dietary adequacy of Vitamin D and Calcium” were underestimated. I suggest the inclusion of these variables. Presenting and discussing only the use of supplements (obvious results) eliminates the possibility of the study contributing to dietary intervention actions in the population.

Figure 1 can be deleted from the manuscript. I suggest maintaining only presenting the data in the table.

§  Discussion: 

I suggest exclusion in the face of repetition of results and bibliographic review style. Include a discussion of diet (adequacy of Vit D and Ca, yes or not?), including concerning significant calories. If there’s is a specification on supplements, it's recommendable to discuss it too.

“In our population, oily fish consumption was low and participants did not consume cereals fortified with calcium and vitamin D. In addition, some participants consumed less or only skimmed milk”. Which study are you referring to?

Author Response

thanks, please see the attachment in the box 

Round 2

Reviewer 1 Report

No comments.